

# The effect of peer support on HbA1c levels in middle-aged and elderly patients with type 2 diabetes: a systematic review and meta-analysis

Jiankun Luo[1,*], Jiyuan Shu[1,*], Weiqing Pan[1], Duanqin Guan[1], Congmin Liang[1], Dongyan Zheng[1], Kailun Huang[1], Yixi Lai[1], Bing Meng[2], Renjian Lu[1], Ziwei Cai[1], Lifei Xing[3], Jialong Chen[1] and He Zhang[1]

[1] Department of Environmental and Occupational Health, School of Public Health, Guangdong Medical University, Dongguan, China
[2] Dongguan Shipai Hospital, Donguan, Guangdong, China
[3] Dongguan First Hospital Affiliated to Guangdong Medical University, Dongguan, Guangdong Province, China
* These authors contributed equally to this work.

Corresponding authors
Jialong Chen, chenjialongaa@163.com
He Zhang, zhangh@gdmu.edu.cn

## ABSTRACT

**Background**. Peer support is increasingly recognized as a valuable method for managing blood glucose levels and reducing the risk of chronic complications in patients with diabetes. This systematic review and meta-analysis specifically evaluated the effect of peer support on HbA1c levels in middle-aged and elderly patients with type 2 diabetes and explored the potential benefits of peer interactions on glycemic control.
**Methodology**. A comprehensive search was conducted across six databases from January 2018 to July 2023, with a focus on randomized controlled trials (RCTs) that compared peer support interventions to standard diabetic care in adults. Among the 3,395 articles identified, eleven studies involving 2,187 participants were included. The quality of each study was assessed using the Cochrane risk of bias tool. A random-effects model was employed to calculate the mean difference (MD) in HbA1c changes, with additional subgroup analyses for specific contexts.
**Results**. The findings revealed a modest but statistically significant improvement in HbA1c levels in groups receiving peer support (MD: $-0.20$; 95% CI [$-0.37$ to $-0.02$]; $p = 0.03$). More pronounced benefits were observed in interventions conducted within formal medical settings, those involving high-frequency contact, and those in group sessions, particularly among elderly patients.
**Conclusions**. The results suggest that while peer support provides a slight improvement in HbA1c levels, other intervention strategies—such as frequent contact, group sessions, and formal medical settings—may offer greater glycemic control benefits in elderly patients with type 2 diabetes. These findings support incorporating peer support into diabetes care and underscore the need for further research with larger samples and standardized protocols.

## INTRODUCTION

The global increase in type 2 diabetes, driven by aging populations and urbanization, has become a major public health issue (*American Diabetes Association, 2020*). According to the International Diabetes Federation (IDF) and Global Burden of Disease data, in 2021, there were already over 530 million people living with diabetes worldwide—approximately 90% of whom had type 2 diabetes (*Bloomgarden, 2024*; *Schnell et al., 2022*). Current estimates suggest that this number will continue to increase, reaching 643 million by 2030 (*Guariguata et al., 2014*; *Saeedi et al., 2019*). This burden not only decreases individuals' quality of life, but it also imposes a significant strain on health care systems, with its global economic impact projected to reach 2.2% of GDP by 2030 (*Bommer et al., 2018*).

The existing standards of care in diabetes management primarily focus on pharmacotherapy, dietary modification, and lifestyle changes (*Correia et al., 2019*; *Kalra, Jena & Yeravdekar, 2018*). However, these approaches sometimes prove insufficient owing to poor patient adherence, making the case for alternative, complementary strategies. Peer support, in the context of health care, refers to the provision of emotional, appraisal, and informational assistance (*Werner, Ufholz & Yamajala, 2024*). This support is offered by an individual who is part of a constructed social network and possesses experiential knowledge pertinent to a specific behavior or stressor. This individual also shares demographic or experiential characteristics with the target population (*Horne et al., 2023*). The purpose of such support is to address health-related concerns affecting individuals who are experiencing, or are at risk of experiencing, psychological or physical stress (*Mikolajczak-Degrauwe et al., 2023*).

Several empirical studies have begun to elucidate the role of peer support in improving diabetes outcomes. For example, *Fisher et al. (2015)* and *Shalaby & Agyapong (2020)* reported that peer support has the potential to enhance self-management and glycemic control. While pharmacotherapy remains the cornerstone of diabetes management, the integration of peer support can provide a more holistic approach, offering emotional, social, and practical support for daily disease management.

Nevertheless, the application of peer support is not devoid of limitations. Although peers can offer invaluable emotional support and experiential knowledge, they are generally not qualified to provide medical advice or adjust treatment regimens (*Coakley et al., 2023*; *Gerritzen, McDermott & Orrell, 2022*; *Horne et al., 2021*; *Wells et al., 2022*). Furthermore, the effectiveness of peer support is not universally consistent, as it may be contingent upon individual predilections for social interaction and the quality of the peer relationship (*Brasier et al., 2022*; *Crompton et al., 2022*).

Despite the accumulating evidence on the effectiveness of peer support in existing academic research, a specific gap remains in the current literature. Most existing meta-analyses and systematic reviews, such as those by *Patil et al. (2016)* and *Azmiardi et al. (2021)*, have broadly explored the efficacy of peer support interventions but have left the impact of specific peer and patient characteristics on HbA1c outcomes under investigation.

To address this research gap, the present systematic review and meta-analysis aimed to focus explicitly on the effect of peer support on hemoglobin A1c (HbA1c) levels in

middle-aged and elderly patients with type 2 diabetes. Our primary objective was to assess the overall effectiveness of peer support for glycemic control. Furthermore, we aimed to examine the underlying factors that contribute to this effectiveness, with a particular focus on how peer and patient characteristics such as age, duration of diabetes diagnosis, and level of training influence these outcomes.

# SURVEY METHODOLOGY

## Search strategy

This meta-analysis followed the Preferred Reporting Items for Systematic Reviews and Meta-Analyses (PRISMA) statement (*Moher et al., 2009*). Randomized controlled trials (RCTS) were searched in Cochrane library, PubMed, EMbase, China Biology Medicine Disc (CBM), China National Knowledge Infrastructure (CNKI), Wanfang Database and VIP database by using subject words and keywords, supplemented by manual retrieval. An initial search was performed based on the participants, comparison, intervention, and outcomes (PICO) framework and key terms. We also conducted an additional search by screening the reference lists from the selected literature. The search terms and search strategy are shown in Table S1.

## Inclusion and exclusion criteria

In this study, we included RCTs that compared peer support intervention with otherwise similar care in adults with diabetes that measured HbA1c level as a primary or secondary outcome.

The inclusion criteria were randomized controlled trials (RCTs) involving middle-aged and elderly patients with type 2 diabetes mellitus (T2DM), defined as adults aged over 40 years based on definitions of middle age provided by the Encyclopaedia Britannica and Wikipedia. This lower age cut-off was selected to ensure comprehensive inclusion of individuals in the early transitional phase of middle age who are at risk for type 2 diabetes. Participants were also required to be in good mental health (defined as the absence of diagnosed psychiatric disorders or cognitive impairments, as reported by the primary study authors) and capable of self-care. Patients with severe diabetes complications, other serious heart, brain, lung and kidney diseases, such as stroke, myocardial infarction, malignant tumors and gestational diabetes mellitus were excluded. If the control group was given routine nursing intervention, and the experimental group was given peer support intervention (including telephone follow-up, group teaching, peer discussion, *etc.*), and the diabetes with comprehensive health education knowledge, good communication, and voluntary dissemination of type 2 diabetes health knowledge were selected as peer supporters, we included these studies. All included RCTs reported obtaining ethical approval and informed consent from participants, consistent with standard research practices for randomized controlled trials.

## Extraction of data and study selection

Two researchers (JL, HZ) independently screened the title and abstract of the literature, and then further read the literature to extract the correct data if it met the inclusion

and exclusion criteria. In the event of disagreements between two researchers, a third party was invited to participate in the discussion to facilitate resolution. The individuals responsible for arbitrating these differences were Jialong Chen and He Zhang. Utilizing their expertise and impartial perspective, they guided the team through a collaborative process aimed at reconciling viewpoints and reaching a consensus. This approach ensured that the decision-making process was robust, transparent, and in line with the best practices of scientific inquiry. The extraction content included: ① Basic information of the included literature: the first author, the title and the time; ② Basic characteristics of the subjects: sample size, age, gender, *etc.*; ③ Enrollment criteria for peers and training for peers intervention measures; ④ Number of contact hours per session, duration of the intervention; ⑤ Frequency of contact, mode of delivery; ⑥ Control measures; ⑦ Research design; ⑧ Primary outcome measures. Two authors independently (Luo J and Zhang H) extracted data from the included articles into a structured table. Study characteristics are shown in Table S2.

## Bias and quality assessment

The quality of the included studies was evaluated by two researchers using the Cochrane Collaboration's Risk of Bias tool. The assessment tool includes seven items: Methods of randomization (selection bias), concealment of allocation (selection bias), blinding of patients and trial personnel (implementation bias), blinding of outcome assessors (implementation bias), incomplete outcome data (follow-up bias), selective reporting (reporting bias), and other sources of bias were excluded. The risk of bias for each domain was categorized as high, moderate, low (*Higgins et al., 2011*).

## Design

This review was registered with the International Prospective Systematic Review Registry (PROSPERO) under the registration number CRD42023462231. The PRISMA checklist for systematic reviews and meta-analyses is also attached to Table S3 of this review.

## Data synthesis and statistical analysis

Our meta-analysis was executed leveraging Review Manager 5.4 and Stata version 17 to ensure rigorous and comprehensive data synthesis. When assessing outcomes standardized on equivalent scales, we calculated the mean difference (MD) supplemented by its 95% confidence interval (95% CI). In instances where studies omitted standard deviations (SDs), we employed established statistical methods: SDs were inferred from the available data, extrapolated from $P$ values and CIs in alignment with the Cochrane Collaboration's guidelines, or imputed drawing from baseline metrics (*Higgins et al., 2024*; *Wan et al., 2014*).

The primary outcome, consistent with the established markers of diabetes management, centered on the variation in HbA1c levels from baseline to the study endpoint. We meticulously computed the mean difference between initial and concluding HbA1c levels for both intervention and control cohorts, ensuring the precision of the accompanying SD for each computed difference. This yielded a pooled MD in HbA1c fluctuations, bolstered by a 95% CI (*DerSimonian & Laird, 1986*).

To validate the consistency across studies, we harnessed Cochran's Q and $I^2$ statistics. We identified significant heterogeneity at either a $P$-value threshold of $< 0.10$ or an $I^2$ statistic surpassing 50%. Given the inherent differences in study populations, intervention settings, and peer support modalities across the included studies, we adopted a random-effects model for all analyses. This approach aligns with methodological guidance in the Cochrane Handbook and recent meta-analyses in the field, which recommend using a random-effects model to account for between-study heterogeneity in complex behavioral interventions (*Higgins et al., 2024*). To address potential publication bias, we constructed funnel plots and conducted Egger's test. A $p$-value of less than 0.10 in the Egger's test was interpreted as indicative of publication bias. Additionally, we performed a leave-one-out sensitivity analysis, sequentially omitting one study at a time and recalculating the pooled mean difference (MD). This approach allowed us to assess whether any single study disproportionately influenced the overall effect size, thereby confirming the robustness of our meta-analytic results (*Egger et al., 1997*; *Godavitarne et al., 2018*).

Recognizing the diverse facets that influence diabetes outcomes, we incorporated subgroup analyses. This stratification was rooted in factors demonstrably impactful in diabetes care and research: participant age (categorized as >60 years or ≤60 years), intervention duration (either ≤6 months or extended >6 months), the intervention setting (distinguishing formal medical institutions from informal care settings), the modality of intervention delivery (ranging from individualized sessions, group sessions, to hybrid approaches), and the consultation frequency (quantified as low, moderate, or high based on monthly contacts).

## RESULTS

### Study selection

Figure 1 shows the PRISMA flowchart of the article selection process. A total of 3,395 references were retrieved, 288 duplicate references were removed, 1,955 references were excluded if their duration did not meet the requirements, 1,053 references were excluded after carefully reviewing the title and abstract, and 88 articles were excluded according to the inclusion and exclusion criteria after reading the full text of the articles during the second screening of the literature. Finally, 11 articles were included in this study (*Andreae et al., 2021*; *Castillo-Hernandez et al., 2021*; *Chen et al., 2021*; *Debussche et al., 2018*; *Ju et al., 2018*; *Long et al., 2020*; *Peimani et al., 2018*; *Pienaar, Reid & Nel, 2021*; *Presley et al., 2020*; *Tang et al., 2022*; *Zhao, Yu & Zhang, 2019*).

### Quality of the included studies

The results of the risk of bias analysis are shown in Fig. 2. Overall, of the 11 included studies, eight studies adequately described randomization sequence generation. Six studies did not describe allocation concealment, whereas six studies did not explain the blinding of personnel and participants, and one of these studies described a high risk of bias. Six studies did not describe blinding to the outcome assessment. There was a low risk of attrition bias due to incomplete outcome data in only one study. All 11 studies had a low risk of bias for selective reporting and other bias. With respect to statistical power, most

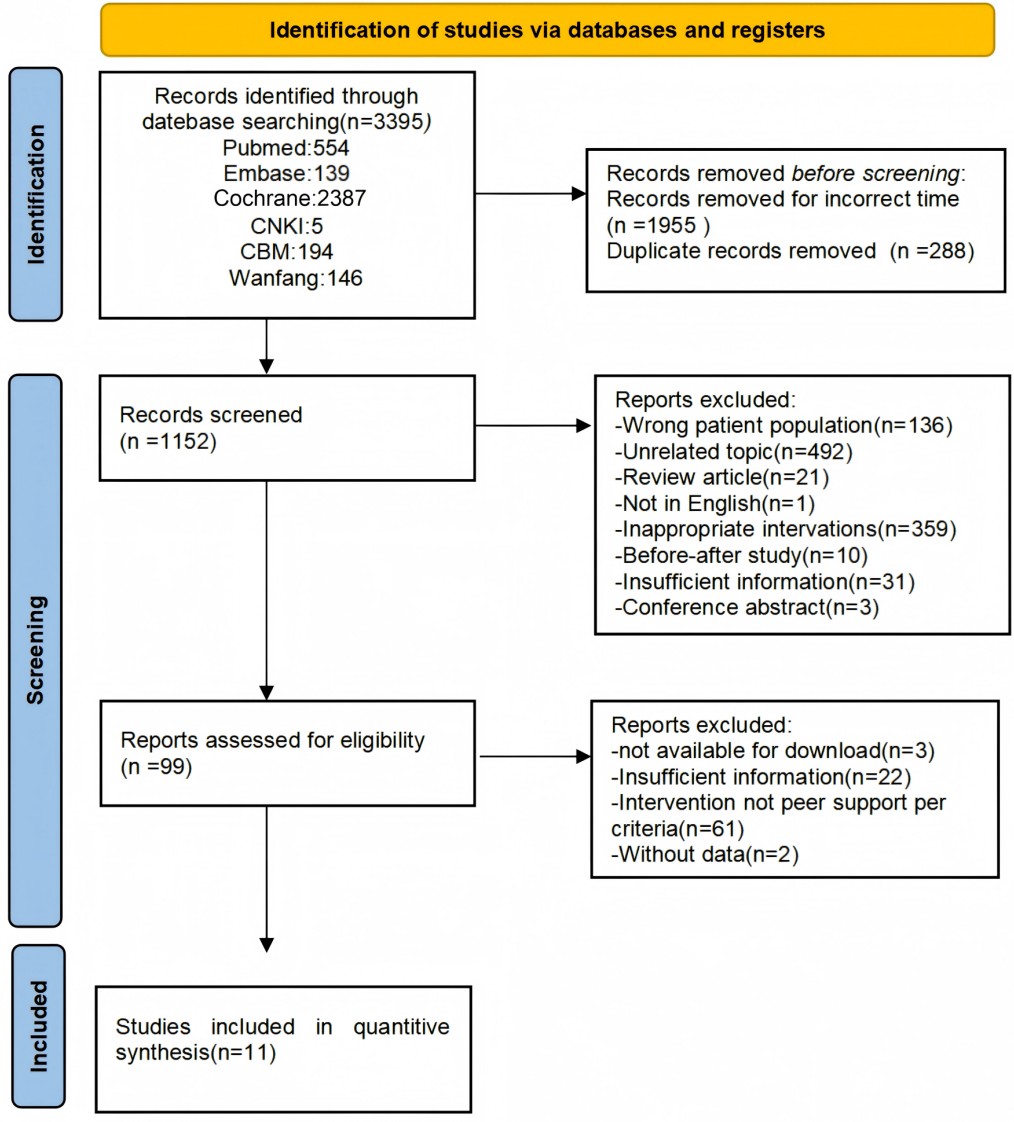

**Figure 1 PRISMA flow diagram.** The process of study identification, screening, eligibility assessment, and inclusion in a systematic review and meta-analysis of the effects of peer support on HbA1c levels in patients with type 2 diabetes. The numbers at each stage represent the count of studies.

studies reported formal power and sample size calculations, typically using an 80% power threshold. However, three studies—*Chen et al. (2021)*, *Presley et al. (2020)*, and *Zhao, Yu & Zhang (2019)*—did not report power or sample size calculations. The overall evidence was originally evaluated as high quality. However, owing to the heterogeneity of the study findings, the quality of evidence was downgraded from high to moderate.

## Impact of peer support on HbA1c levels

Figure 3 shows the effect size and 95% CIs of the included studies. Eleven studies evaluated the effects of peer support teaching and routine nursing education on HbAlc levels in

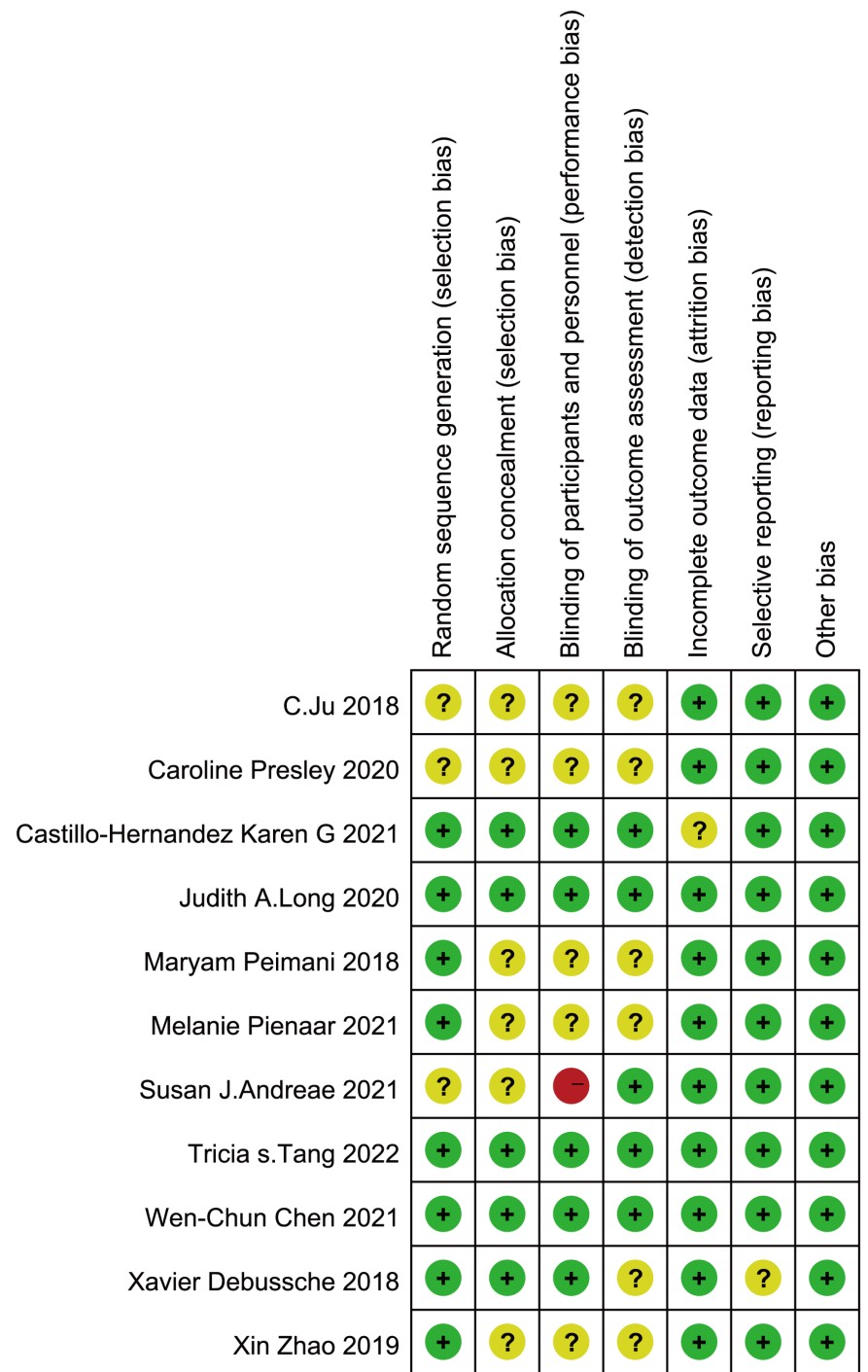

**Figure 2 Risk of bias summary: systematic review of peer support on Type 2 Diabetes management.**
The risk of bias assessment for each included study in the systematic review. Green plus signs (+) indicate low risk of bias, yellow question marks (?) indicate unclear risk of bias, and red minus signs (-) indicate high risk of bias across different domains of the study design. Note: *Ju et al. (2018)*; *Presley et al. (2020)*; *Castillo-Hernandez et al. (2021)*; *Long et al. (2020)*; *Peimani et al. (2018)*; *Pienaar, Reid & Nel (2021)*; *Andreae et al. (2021)*; *Tang et al. (2022)*; *Chen et al. (2021)*; *Debussche et al. (2018)*; *Zhao, Yu & Zhang (2019)*.

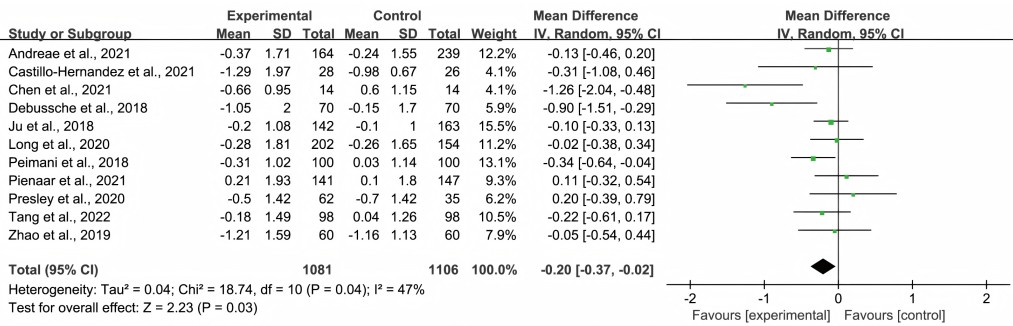

**Figure 3** **Forest plot: impact of peer support on HbA1c levels in Type 2 Diabetes patients.** Each study is represented by a square that corresponds to the study's estimate of the effect size, with the size of the square reflecting the study's weight in the meta-analysis. The horizontal line through each square represents the 95% confidence interval. The diamond at the bottom of the plot represents the overall effect size, with its width indicating the 95% confidence interval for the pooled estimate. Note: *Ju et al. (2018)*; *Presley et al. (2020)*; *Castillo-Hernandez et al. (2021)*; *Long et al. (2020)*; *Peimani et al. (2018)*; *Pienaar, Reid & Nel (2021)*; *Andreae et al. (2021)*; *Tang et al. (2022)*; *Chen et al. (2021)*; *Debussche et al. (2018)*; *Zhao, Yu & Zhang (2019)*.

patients with T2DM. Because of the high heterogeneity of the studies, a random effects model was employed to evaluate possible differences in HbA1c levels between the control and intervention groups. The overall pooled effect of peer support interventions on HbA1c levels was an MD of −0.20 (95% CI [−0.37 to −0.02]; $p = 0.03$), and it was statistically significant, favoring peer support over usual care. High statistical heterogeneity was found among the studies in terms of changes in HbA1c levels ($I^2 = 47\%$, $p = 0.04$).

## Subgroup analyses

Subgroup analyses were executed post hoc to discern potential variations attributed to the distinct characteristics intrinsic to each study, with specific findings detailed in Table 1. (See Figs. S1, S2, S3, S4, and S5).

Our findings delineate differential responses in HbA1c reduction contingent upon participant age. Compared with their younger counterparts (MD −0.20 [−0.37 to −0.02]), participants older than 60 years exhibited a more pronounced reduction (MD −0.30 [−0.85 to 0.25]) in HbA1c levels.

The institutional context of intervention delivery was a significant determinant of outcome variability. Interventions administered within formal medical institutions were associated with an amplified reduction in HbA1c levels (MD −0.32 [−0.72 to 0.08]) relative to those orchestrated in informal settings (MD −0.16 [−0.30 to −0.01]).

The modality of educational delivery also affected HbA1c outcomes. Participants who engaged in group sessions or experienced a hybrid of group and individual sessions manifested a discernible decline in HbA1c levels (MD −0.37 [−0.67 to −0.06] and MD −0.17 [−0.60 to 0.26], respectively). In contrast, exclusively individualized educational interventions did not yield a statistically significant decrease (MD −0.12 [−0.33 to 0.09]).

**Table 1  Subgroup analysis.**

| Study characteristics | Number of studies/ participants | HbA1c MD (95% CI) | p-value | Heterogeneity | |
|---|---|---|---|---|---|
| | | | | **P** | **I2 (%)** |
| **All studies** | 11/2,187 | −0.20 [−0.37, −0.02] | .03 | .04 | 47% |
| **Age of patients** | | | | | |
| ≤60 | 8/1,566 | −0.20 [−0.37, −0.02] | .03 | .24 | 24% |
| >60 | 3/621 | −0.30 [−0.85, 0.25] | .29 | .01 | 78% |
| **Duration** | | | | | |
| ≤6 mo | 6/1,136 | −0.18 [−0.47, 0.10] | .21 | .03 | 59% |
| >6 mo | 5/1,051 | −0.22 [−0.45, 0.02] | .07 | .16 | 39% |
| **Type of peer support** | | | | | |
| Individual | 4/514 | −0.12 [−0.33, 0.09] | .25 | .76 | 0% |
| Group | 3/955 | −0.37 [−0.67, −0.06] | .02 | .21 | 33% |
| Both | 4/718 | −0.17 [−0.60, 0.26] | .44 | .02 | 71% |
| **Frequency of contact** | | | | | |
| Moderate | 6/1,546 | −0.03 [−0.18, 0.12] | .68 | .81 | 0% |
| High | 5/710 | −0.32 [−0.55, −0.01] | .005 | .3 | 17% |
| **Location** | | | | | |
| Formal medical institutions | 5/1,177 | −0.16 [−0.30, −0.01] | .04 | .65 | 0% |
| Informal medical institutions | 6/1,010 | −0.32 [−0.72, 0.08] | .12 | .007 | 69% |

With regard to contact frequency, higher-frequency contacts were more efficacious (MD −0.32 [−0.55 to −0.1]) in mitigating HbA1c levels than moderate-frequency contacts were (MD −0.03 [−0.18 to 0.12]).

In evaluating the temporal dimension of interventions, our analysis revealed no substantial differential outcomes between shorter-term interventions (≤6 months, MD −0.18 [−0.47 to 0.1]) and their longer-term (>6 months) counterparts (MD −0.22 [−0.45 to 0.02]) in terms of comparative efficacy between the peer support cohort and the control cohort.

## Publication bias and sensitivity analyses

We explored the possibility of publication bias for the 11 included studies. Sensitivity analyses were performed to test the robustness of the results. In the sensitivity analyses, the pooled results for HbA1c levels did not significantly change after any single study was excluded. There was no publication bias based on Egger's test ($p = 0.24$) or the shape of the funnel plot (Fig. S6).

## DISCUSSION

The focus of this systematic review and meta-analysis was to evaluate the effectiveness of peer support in mitigating glycated hemoglobin (HbA1c) levels among middle-aged and elderly individuals with type 2 diabetes in recent years. The data suggest that peer support, particularly support provided by diabetic peers, instigates a modest yet statistically

significant reduction in HbA1c levels by 0.20% (95% CI [−0.37% to −0.02%]) compared with standard care education. This diminutive decline is not merely numerical but also has a palpable clinical impact, as even nuanced enhancements in glycemic control are known to significantly decelerate the progression to vascular complications (*Ikeda & Shimazawa, 2019*; *Qaseem et al., 2018*).

This analysis delineates a conspicuous differential outcome in the efficacy of peer support interventions on the basis of the health care setting. Notably, the elevated efficacy observed in formal health care settings, as opposed to informal contexts, is notable. This discrepancy potentially stems from the inherent structured and professionally overseen environment in formal settings, which might propagate a more rigorous adherence to intervention strategies than informal settings, which are typically characterized by autonomous lifestyle choices and familiar environments (*Egbujie et al., 2018*; *Lee et al., 2017*). The dissected data further offer insights into the correlated efficacy and frequency of peer interventions. High-frequency peer support materializes as a tangible tool for enhancing glycemic control, whereas medium-frequency interventions do not substantively sway HbA1c levels. This alignment with prior meta-analyses suggests an imperative to develop diabetes management strategies, particularly for middle-aged and elderly individuals, with high-frequency peer support. A deeper exploration into the age-dependent efficacy of peer support reveals a pronouncedly beneficial impact among older patients. Here, the intertwining of emotional and medical support might facilitate an enriched environment for managing diabetic outcomes, given that the emotional reassurance offered by peers can potentially mitigate prevalent negative emotions among older adults.

This study has limitations. A significant obstacle encountered was the pronounced heterogeneity across the 11 included randomized controlled trials, notably in peer recruitment and training protocols. The frequent omission of peers' basic characteristics and a lack of uniformity in training methods necessitated the deployment of a random-effects model and circumscribed the granularity of subgroup analyses. This highlights the need for future research to be underpinned by detailed and standardized reporting on peer and participant characteristics, thereby facilitating a more detailed meta-analytic review in subsequent studies.

## CONCLUSIONS

Conclusively, this meta-analysis revealed that peer support emerges as a vital agent in modulating HbA1c levels among middle-aged and elderly individuals with type 2 diabetes, outpacing the efficacy of standard nursing education. Notably, the implementation of peer interventions involving formal medical institutions, group sessions, and frequent sessions has been found to significantly improve glycemic control in elderly patients with type 2 diabetes mellitus. Health care systems must recognize the significant potential of peer support networks and judiciously recruit individuals endowed with experiential knowledge of diabetes management. Further research, marked by larger sample sizes and in-depth

participant characteristics, will amplify the precision of these findings and enrich the foundation for clinical decision-making (*American Diabetes Association, 2020*).

### Funding

This study was funded by the National Natural Science Foundation of China (82103879), GuangDong Basic and Applied Basic Research Foundation (2021B1515140032, 2022A1515140173, 2023A1515140164), Guangdong Medical University Clinical and Basic Technology Innovation Special Program (GDMULCJC2024152, GDMULCJC2024153, GDMULCJC140) and Dongguan Social Development Science and Technology Key Project (20211800905332). The funders had no role in study design, data collection and analysis, decision to publish, or preparation of the manuscript.

### Grant Disclosures

The following grant information was disclosed by the authors:
The National Natural Science Foundation of China: 82103879.
GuangDong Basic and Applied Basic Research Foundation: 2021B1515140032, 2022A1515140173, 2023A1515140164.
Guangdong Medical University Clinical and Basic Technology Innovation Special Program: GDMULCJC2024152, GDMULCJC2024153, GDMULCJC140.
Dongguan Social Development Science and Technology Key Project: 20211800905332.

### Competing Interests

The authors declare there are no competing interests.

### Author Contributions

- Jiankun Luo performed the experiments, prepared figures and/or tables, authored or reviewed drafts of the article, and approved the final draft.
- Jiyuan Shu analyzed the data, authored or reviewed drafts of the article, and approved the final draft.
- Weiqing Pan performed the experiments, authored or reviewed drafts of the article, and approved the final draft.
- Duanqin Guan performed the experiments, prepared figures and/or tables, and approved the final draft.
- Congmin Liang performed the experiments, authored or reviewed drafts of the article, and approved the final draft.
- Dongyan Zheng conceived and designed the experiments, analyzed the data, authored or reviewed drafts of the article, and approved the final draft.
- Kailun Huang analyzed the data, authored or reviewed drafts of the article, and approved the final draft.
- Yixi Lai analyzed the data, authored or reviewed drafts of the article, and approved the final draft.

- Bing Meng conceived and designed the experiments, prepared figures and/or tables, and approved the final draft.
- Renjian Lu analyzed the data, authored or reviewed drafts of the article, and approved the final draft.
- Ziwei Cai performed the experiments, prepared figures and/or tables, authored or reviewed drafts of the article, and approved the final draft.
- Lifei Xing performed the experiments, prepared figures and/or tables, and approved the final draft.
- Jialong Chen conceived and designed the experiments, authored or reviewed drafts of the article, and approved the final draft.
- He Zhang conceived and designed the experiments, authored or reviewed drafts of the article, and approved the final draft.

## Data Availability

Data is available in the Supplemental Files.

## Supplemental Information

Supplemental information for this article can be found online at http://dx.doi.org/10.7717/peerj.19803#supplemental-information.

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
