# Peer review of "The effect of peer support on HbA1c levels in middle-aged and elderly patients with type 2 diabetes: a systematic review and meta-analysis"

_PeerJ, doi:10.7717/peerj.19803_

## Round 0.1 · original submission · Minor Revisions

Dear Dr. Chen,

Your manuscript entitled “The effect of peer support on HbA1c levels in middle-aged and elderly patients with type 2 diabetes: a systematic review and meta-analysis", which you submitted to PeerJ, has been reviewed by the editor and 3 experts in the field.

The reviewers are generally favorable and suggest that, subject to minor revisions, your paper could be suitable for publication. Please consider these points carefully, as the revised manuscript will undergo a second round of review by the same reviewers.

I hope you will be prepared to make the necessary amendments and submit a revised manuscript accompanied by a statement of how you have responded to the reviewers’ comments, particularly those concerning the Experimental design and Validity of the findings from Reviewer 1.

Yours sincerely,

Stefano Menini

**Language Note:** The review process has identified that the English language must be improved. PeerJ can provide language editing services - please contact us at [email protected] for pricing (be sure to provide your manuscript number and title). Alternatively, you should make your own arrangements to improve the language quality and provide details in your response letter. – PeerJ Staff

·

Basic reporting

The abstract clearly outlines the study's aim to evaluate the impact of peer support on HbA1c levels in middle-aged and elderly patients with type 2 diabetes, a timely and relevant topic given the growing interest in non-pharmacological interventions for managing chronic diseases. The study focuses on glycemic control through peer interactions, offering a low-cost solution. The research question is well-defined and important for current clinical practice, especially in managing type 2 diabetes. The methodology, including a systematic review and meta-analysis, is robust, allowing for a comprehensive understanding of the effects of peer support on HbA1c levels.

Experimental design

The study uses a systematic review and meta-analysis of randomized controlled trials (RCTs), which is a robust design for synthesizing evidence and evaluating the impact of peer support on HbA1c levels in type 2 diabetes patients. However, ethical considerations, such as institutional review board approvals and informed consent, should be addressed for the included studies. While the inclusion and exclusion criteria are well-defined, further clarification is needed on aspects like the definition of "good mental state," routine care, and the peer support selection process to enhance transparency. Additionally, providing a rationale for selecting 50 years as the lower age limit for inclusion would improve the study's clarity.

Validity of the findings

The study presents a well-conducted systematic review and meta-analysis with robust data and reliable statistical methods, offering valuable insights into the impact of peer support on HbA1c levels in type 2 diabetes patients. However, it would benefit from a clearer discussion on novelty, impact, and clinical significance, as well as addressing study limitations and potential future research areas.

Additional comments

It would be beneficial to include a clear explanation of how the sample size was determined for this meta-analysis. For instance, was there a pre-specified sample size based on power analysis to detect an effect of peer support on HbA1c levels? If so, it would be helpful to describe the effect size that was anticipated and the desired power (e.g., 80% power) for the study.
Providing the sample size calculation for each individual RCT included in the meta-analysis would offer additional transparency, especially regarding how each study was powered to detect the expected difference in HbA1c levels.

Reviewer 2 ·

Basic reporting

The article written in clear and unambiguous manner. Sufficient field background was provided. Clear figures and table s were provided with both in article and supplementary materials
no comment to add

Experimental design

The authors followed all the rules for systematic review and meta analysis has to follow .The design was written well with clear cut methodology description .
no comments to be added

Validity of the findings

The are clearly justified with clear conclusion. The findings are not only the statistically sound ,also which is important to focus more on that . Due the heterogeneity of different articles may be leaves the incomplete conclusions
no comment to add

Additional comments

no additional comment to add

Reviewer 3 ·

Basic reporting

No comment

Experimental design

No comment

Validity of the findings

No comment

Additional comments

Throughout: Check for typographical and formatting errors. Lns 55, 154,197, 199,

Abstract:
Ln 40: "specific intervention strategies. " It could be termed 'other' intervention strategy because 'peer support' is also a form of intervention strategy.

Introduction
Ln 47: "an estimated 422 million people suffer from diabetes." Is this a global estimate, and what year was this reported?
Ln 55: Be consistent with the paragraph style.
Ln 110: Kindly justify why the standard deviation was applied to the lower cut-off for age.
Ln 154: The specific outcomes reported using SMD should be specified.
Ln 156-159: "In instances where studies omitted standard deviations (SDs), we employed established statistical methods: SDs were inferred from the available data, extrapolated from P values and CIs in alignment with the Cochrane Collaboration's guidelines, or imputed drawing from baseline metrics" Provide citations for this statement, including previous literatures that used similar methods.
Ln 168: Format I-square throughout the manuscript.
Ln 169: Kindly provide or cite relevant sources to justify the threshold selected for heterogeneity.
Ln 169-172: Model selection, generally, statistical tests of heterogeneity should not be used to select a model. I would argue that an understanding of how these studies were sampled supports the use of a random effects model. Would recommend revisiting this and presenting a random effects model for all outcomes or providing justification, including precedence if available, for retaining the strategy of model selection based on heterogeneity statistics.
Ln 175: Kindly provide further clarification about the sensitivity analysis, specifically, the statement "We iteratively excluded individual studies." How were the excluded studies determined?

Ln 154-155: Although the authors highlighted that SMD was used for outcomes derived from heterogeneous scales, none of the final results were reported using SMD. Kindly clarify, else, this should be removed from the methods.

---

## Round 0.2 · accepted · Accept

Dear Dr. Chen and Zhang,

Thank you for submitting the revised version of your manuscript. I have personally reviewed the revision and confirmed that all the reviewers' comments have been adequately addressed. The quality of the manuscript has significantly improved as a result. I am pleased to inform you that your manuscript is now ready for publication in PeerJ in its current form.

Sincerely,
Stefano Menini